# The thermal response of soil microbial methanogenesis decreases in magnitude with changing temperature

Hongyang Chen [1], Ting Zhu [1], Bo Li [1], Changming Fang[1] & Ming Nie [1✉]

Microbial methanogenesis in anaerobic soils contributes greatly to global methane ($CH_4$) release, and understanding its response to temperature is fundamental to predicting the feedback between this potent greenhouse gas and climate change. A compensatory thermal response in microbial activity over time can reduce the response of respiratory carbon (C) release to temperature change, as shown for carbon dioxide ($CO_2$) in aerobic soils. However, whether microbial methanogenesis also shows a compensatory response to temperature change remains unknown. Here, we used anaerobic wetland soils from the Greater Khingan Range and the Tibetan Plateau to investigate how 160 days of experimental warming (+4°C) and cooling (−4°C) affect the thermal response of microbial $CH_4$ respiration and whether these responses correspond to changes in microbial community dynamics. The mass-specific $CH_4$ respiration rates of methanogens decreased with warming and increased with cooling, suggesting that microbial methanogenesis exhibited compensatory responses to temperature changes. Furthermore, changes in the species composition of methanogenic community under warming and cooling largely explained the compensatory response in the soils. The stimulatory effect of climate warming on soil microbe-driven $CH_4$ emissions may thus be smaller than that currently predicted, with important consequences for atmospheric $CH_4$ concentrations.

[1] Ministry of Education Key Laboratory for Biodiversity Science and Ecological Engineering, Coastal Ecosystems Research Station of the Yangtze River Estuary, the Institute of Biodiversity Science, School of Life Sciences, Fudan University, 2005 Songhu Road, Shanghai 200438, China. ✉email: mnie@fudan.edu.cn

Over a 100-year time frame, $CH_4$ has 28 times the global warming potential of $CO_2$, and it is thus expected to play an important role in future climate change[1]. Anaerobic soil $CH_4$ fluxes are a major component of global $CH_4$ emissions[2–4], and short-term experiments have shown that soil microbial methanogenesis is strongly and positively dependent on temperature[5–7]. $CH_4$ cycling simulations with this information indicate that the anaerobic soil $CH_4$ respiration rate is likely to increase sharply as global temperature rises[8,9], triggering a positive climate change-$CH_4$ feedback[8,10,11].

However, the strength of this climate change-$CH_4$ feedback remains uncertain[1], primarily because the response of microbial respiration to long-term temperature change can differ from its instantaneous response[12–14]. In aerobic soils, such as those of forests and grasslands, increasing evidence indicates that a compensatory response of the microbial community can greatly reduce the impact of temperature changes on soil $CO_2$ respiration rates over intermediate to long time scales[15–18]. These adjustments in soil microbial respiration rate responses to temperature changes could result from acclimation (physiological responses of individuals), adaptation (genetic variation within species) and/or species turnover (shifts in the species composition of the community)[15,19–21]. Incorporating such a compensatory response into models improves predictions of the rates of global soil C loss[15,22]. Therefore, given the fundamental effects of temperature on biological metabolism[23,24], it is reasonable to conclude that the compensatory thermal response might be similar in aerobic soils (in terms of $CO_2$ production) and anaerobic soils (in terms of $CH_4$ production). However, until now, no attempt has been made to examine whether microbial methanogenesis shows a compensatory response to temperature change.

To discern whether microbial methanogenesis in anaerobic soils exhibits a compensatory response to temperature change, we collected wetland soil samples from plots established at four sites in the Greater Khingan Range (GKR) and four sites on the Tibetan Plateau (TP) (Supplementary Table 1). Because of the differences in the soil methanogenic community and physicochemical properties between the selected GKR and TP soils[25,26], there might be considerable differences in the thermal responses of methanogens to temperature change in these contrasting soils, and the use of these soils may convincingly test the compensatory response of microbial $CH_4$ respiration to changing temperature and its underlying mechanisms.

## Results and discussion

**Compensatory responses to warming and cooling.** Soil samples from the GKR and the TP were used to establish anaerobic microcosms to evaluate methanogenesis and changes in microbial community composition under experimental warming and cooling. The soil samples were anaerobically preincubated at 12 °C (reference temperature, RT) for 66 days to allow the $CH_4$ respiration rates to be stable[3,20]. The RT was derived from the mean growing-season temperatures in the selected wetlands (see Methods). In general, the compensatory thermal response of the microbial community involves a change in mass-specific respiration ($R_{mass}$) that opposes the effects of the applied change in temperature[17,18,20,23], i.e., $R_{mass}$ should decline following a sustained increase in temperature and rise following a sustained decrease in temperature (Fig. 1a). After the pre-incubation period, we experimentally warmed (RT + 4 °C) and cooled (RT − 4°C) the microcosms for a 160-day incubation (Fig. 1b) (see Methods); this incubation length under such conditions has been hypothesized to allow the compensatory thermal response of microbial respiration to occur[20,27–29].

We conducted short-term assays (Fig. 1b; see Methods) and observed significant warming and cooling effects in both GKR (Fig. 2a) and TP (Fig. 2b) soils (all $P < 0.01$). Specifically, across the range of assay temperatures, $CH_4$-$R_{mass}$ rates in the soils were greatest for the cooling treatment and lowest for the warming treatment (Fig. 2a, b). In addition, the $CH_4$-$R_{mass}$ rate decreased under warming and increased under cooling for both GKR (Fig. 2c) and TP soils (Fig. 2d) at the corresponding assay temperatures. This pattern indicates that microbial methanogenesis exhibited a compensatory response to the temperature changes imposed in our study. A similar pattern was also observed for the thermal response of soil $CH_4$ respiration (not corrected for biomass) to warming and cooling (Supplementary Fig. 1). These findings are consistent with compensatory metabolic responses to altered ambient temperatures that have been observed in aerobic soil microbial respiration[17,18,30]. Furthermore, given that the methanogen biomass did not differ significantly ($P > 0.05$) between the control and treatment soils (Supplementary Fig. 2), the driving force behind this compensatory thermal response may not be the change in microbial biomass[20].

Further analysis showed that in the case of the GKR soils, the magnitude of compensatory responses (MCR) of methanogenesis (see Methods) under experimental warming was significantly higher than that under experimental cooling (Supplementary Fig. 3) ($P < 0.01$), suggesting that the compensatory response of soil microbial $CH_4$ respiration to temperature change in this region may be gradually enhanced by ongoing climate warming. However, this phenomenon was not observed in the TP soils (Supplementary Fig. 3), whose pH (8.0) might impose constraints on their physiological adjustment to rising temperature, as the optimal pH for methanogenic archaea lies between 4 and 7[12,31,32].

The compensatory thermal response of microbial respiration can play an important role in weakening positive soil C-climate feedback[15,20,33], and there are two main types of compensatory thermal responses of $R_{mass}$[18,19,34]: in type I, the temperature sensitivity ($Q_{10}$) of $R_{mass}$ decreases (with no change in $R_{mass}$ at low assay temperatures), while in type II, $R_{mass}$ decreases at both low and high temperatures, without any change in $Q_{10}$ necessarily taking place. Because the overall elevation of the temperature response curve is affected in type II, the degree of weakening of the positive feedback would be greater for type II than for type I[34]. Our results show that there were no significant ($P > 0.05$) differences in the $Q_{10}$ value of $CH_4$-$R_{mass}$ across the three imposed incubation temperatures (Fig. 2a, b and Supplementary Table 2), indicating that the compensatory response of microbial $CH_4$ respiration that we observed was predominantly type II. This finding suggests that future $CH_4$ respiration rates in anaerobic wetland soils may not be as high as currently predicted but would follow the current temperature sensitivity.

**Linking compensatory responses and community structure.** We evaluated whether the observed compensatory thermal response of methanogenesis is related to a change in microbial community dynamics. We observed shifts in the methanogenic community composition under both warming and cooling (Fig. 3a). However, in contrast to the consistent changes in $CH_4$-$R_{mass}$ for both soils (Fig. 2c, d), there was no consistent response to temperature change for the dominant methanogenic archaea between the two soil types. For instance, *Methanotrichaceae* in GKR soils significantly ($P < 0.05$) declined in relative abundance under both experimental warming and cooling, while it significantly increased in relative abundance with increasing temperature in TP soils (Fig. 3b). The relative abundance is referred to the evenness of

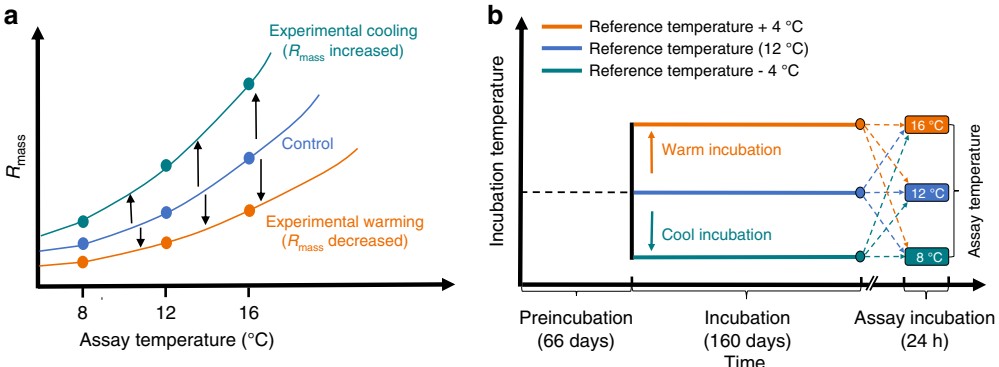

**Fig. 1 Compensatory thermal responses of respiration and the experimental design used to evaluate the responses of soil microbial CH₄ respiration to experimental warming or cooling. a** The patterns of microbial respiration that would be observed in the case of physiological compensatory responses. Respiration rates are expressed per unit of microbial biomass ($R_{mass}$). **b** Our study design. The reference temperature was set at 12 °C to represent the mean growing-season temperature in the selected wetlands. The 66-day anaerobic preincubation period allowed for depletion of inorganic terminal electron acceptors (TEAs) and stabilization of the CH₄ respiration rate. On day 66, three different thermal treatments (experimental warming and cooling and control conditions) were established. At the end of the incubation period, short-term assays were conducted at all three temperatures for each treatment to test for the compensatory response of microbial CH₄ respiration.

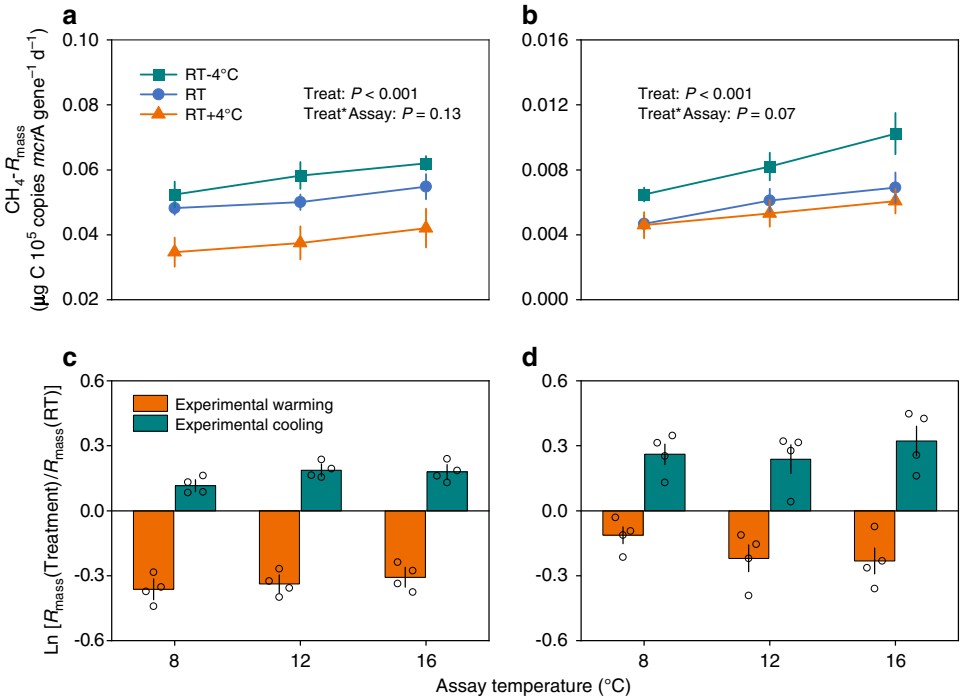

**Fig. 2 The soil mass-specific CH₄ respiration rate decreased under warming and increased under cooling. a**, **c** The Greater Khingan Range; **b**, **d** The Tibetan Plateau. The abundance of *mcrA* gene copies was used as a proxy for the biomass of methanogens. A linear mixed-effects model was used to test for incubation and assay temperature effects on the soil mass-specific CH₄ respiration (CH₄-$R_{mass}$), the fixed factors were the incubation temperature (RT − 4 °C, RT, and RT + 4 °C; Treat) and assay temperature (8, 12, and 16 °C; Assay), and the random factor was the spatial replicates within each wetland in **a** and **b**. For clarity the values in **c** and **d** are the natural log of the treatment (RT − 4 °C or RT + 4°C): control (RT) ratios of CH₄-$R_{mass}$ following 160 days of experimental warming and cooling. Ratios > 0 indicate a greater CH₄-$R_{mass}$ in the treatment than in the control, and ratios < 0 indicate the reverse. RT, reference temperature (12 °C). Data are presented as mean values ± SEM, $n = 4$ independent soil samples.

distribution of individuals among species in a community, and the observed differences in relative abundance may not accurately reflect the quantity of the microbial community and the intersample differences between taxa[35,36]. These may be the reasons why the shifts in the relative abundance of specific methanogens could not explain the compensatory response of microbial methanogenesis to temperature change.

To further understand whether differences in species compositional turnover among the methanogenic communities (β-diversity) were associated with compensatory thermal responses of microbial methanogenesis, we applied a linear mixed-effects model—specifically, we investigated the relationships between the MCR and β-diversity across temperature treatments. We found that the magnitude of the compensatory responses of methanogenesis was positively related to the β-diversity of the methanogenic community under experimental warming and cooling for both GKR ($R^2 = 0.53$, $P = 0.014$) and TP ($R^2 = 0.63$, $P < 0.01$) soils (Fig. 4).

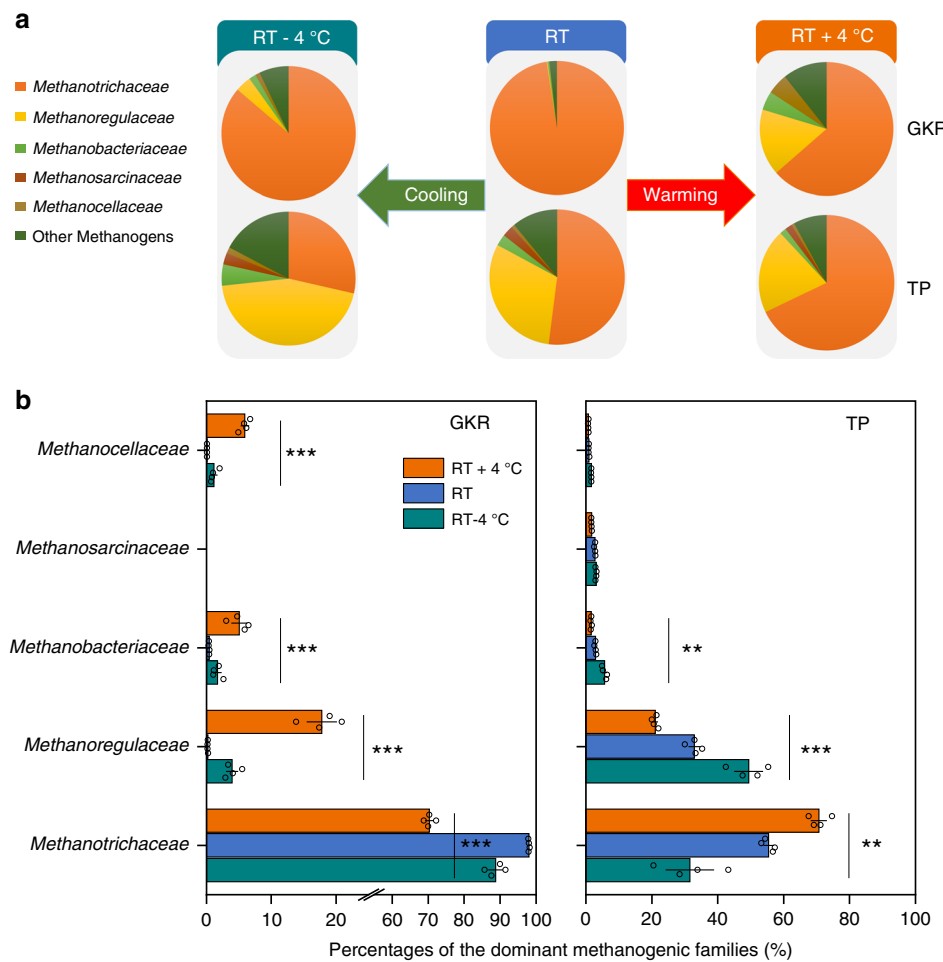

**Fig. 3 Changes in methanogenic community. a** Shifts in methanogenic community composition under experimental warming or cooling by 4 °C. **b** Mean percentages of the dominant methanogenic archaea and their differences among the studied thermal treatments. RT, reference temperature (12°C). GKR, the Greater Khingan Range; TP, the Tibetan Plateau. **$P < 0.001$; ***$P < 0.0001$. Ordinary one-way ANOVA for comparing the three thermal treatments. Data are presented as mean values ± SEM in **b**, $n = 4$ independent soil samples.

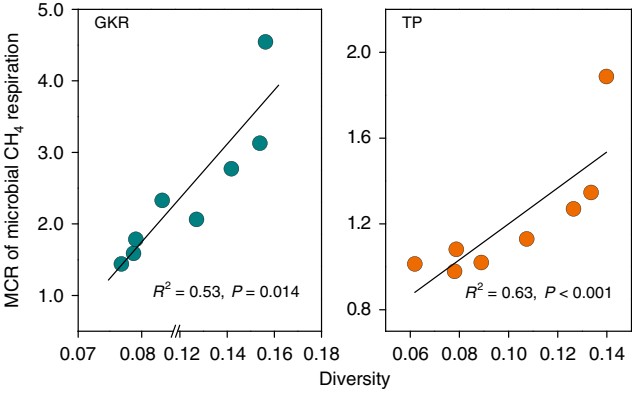

**Fig. 4 Compensatory thermal responses in relation to shifts in the microbial community structure.** The relationship between the magnitude of compensatory responses (MCR) of microbial $CH_4$ respiration and the β-diversity of the methanogenic community was tested by a linear mixed-effects model with thermal treatment (i.e., warming or cooling by 4 °C) as a random effect. GKR, the Great Khingan Range; TP, the Tibetan Plateau.

Changes in community structure are often considered as the key mechanism by which plant communities maintain their functioning under the changing external environment[37,38]. Likewise, we observed that the shifts in community composition of

methanogens were positively associated with the magnitude of their compensatory thermal responses (Fig. 4), reducing the extent to which $CH_4$ respiration rates respond to temperature change. Many models implicitly assume that changes in the community structure of microbial functional groups do not affect soil biogeochemical processes[9,22,39,40]. However, our findings suggest that changes in the methanogenic community structure might be responsible for the compensatory responses of microbial methanogenesis to temperature change, being inconsistent with previous studies of weak linkages between shifts in microbial community composition and the thermal response of microbial $CO_2$ respiration with changing temperature[20,41]. Our findings highlight the current challenge of simulating microbial processes that are carried out by different microbial functional groups[9,39,40,42,43]. They may also help bridge the gap in our understanding of the relationship between microbial community structure and functioning, allowing us to better understand the response of soil $CH_4$ respiration to temperature change in relation to microbial physiology and eventually better predict soil $CH_4$ flux patterns under future climate warming.

In conclusion, our findings provide strong empirical support for the idea that microbial $CH_4$ respiration in anaerobic soils shows a compensatory response to temperature change, indicating that physiologically compensatory response of respiration to the ambient thermal environment may be a common property of microbes in both aerobic and anaerobic soils. In addition, these

findings emphasize that microbial community dynamics plays a vital role in compensating for the thermal response of methanogenesis. In particular, our results imply that the stimulatory effect of climate warming on soil microbe-driven $CH_4$ emissions may be lower than that currently predicted, with important consequences for atmospheric $CH_4$ concentrations. It should be noted that our finding that the compensatory response was predominantly of type II does not suggest that no other type of microbial thermal compensation takes place in anaerobic soils – previous studies have indicated that the types of microbial compensatory adaptation occurring in aerobic soils may be related to both ecosystem type and sampling season[18,29]. Therefore, to gain a better understanding of the microbial control of $CH_4$ emissions at the ecosystem level in a warmer world, future work should assess the potential effects of the soil environment and other biotic factors on the type and MCR of microbial $CH_4$ respiration over larger spatial and temporal scales.

## Methods

**Study soils**. Wetland soil samples were collected from four sites (latitude: 52°25′N to 53°21′N; elevation: 350 to 500 m) in the GKR and four sites (latitude: 37°06′N to 37°42′N; elevation: 3100 to 3400 m) on the TP (Supplementary Table 1). We conducted soil sampling in June and July 2016. At each site, we established a 20 m × 20 m square plot and collected 3 surface (0–20 cm) soil cores at each corner and in the center along a diagonal line. The soils were homogenized by sieving to produce a composite sample. One part was stored at 4 °C for soil anaerobic incubation, and the rest was air-dried for chemical analyses.

**Experimental incubation procedures**. Anaerobic microcosms were prepared by combining 10 g of homogenized soil with 20 ml of deoxygenated water (maintaining a 1:2 soil:water ratio) in autoclaved glass bottles (125 ml). The bottles were then sealed with a cap containing a butyl rubber stopper[44]. During the experiment, the bottles were periodically flushed with high-purity $N_2$ gas to minimize the inhibitory effects of high $CO_2$ and $CH_4$ concentrations on microbial $CH_4$ respiration.

$CH_4$ release from the selected wetlands occurs mainly during the growing season[8], which refers to the period from May to September (when the monthly mean minimum temperature is above 0 °C). The mean growing-season temperatures of the selected GKR and TP wetlands are 13.2 °C and 11.5 °C (1981–2010, climate data from the National Meteorological Information Center of China), respectively (Supplementary Table 1). Therefore, we used the average (12 °C) of the mean growing-season temperature of the two wetlands as the RT in this study (Fig. 1b).

During anaerobic organic matter decomposition, the approximately sequential reduction of inorganic terminal electron acceptors (TEAs) occurs in the order $NO_3^-$, Fe(III), and $SO_4^{2-}$ before $CH_4$ production becomes the sole process[3]. Therefore, all microcosms were subjected to 66 days of preincubation to allow the $CH_4$ respiration rates to completely stabilize[20] with the depletion of inorganic TEAs[3] (Supplementary Fig. 4). At the start of the preincubation period, all microcosm bottles were placed in a thermostatic bath at RT ( ± 0.1 °C). On day 66, the bottles were randomly divided into three equivalent aliquots; two aliquots were transferred to separate thermostatic baths set at RT + 4 °C and RT − 4 °C, and the other aliquot was maintained in the thermostatic bath at RT (Fig. 1b). A total of 336 bottles (2 wetlands × 4 sites × 3 thermal treatments × 14 soil replicates) were used for the 160-day main incubation period. The 14 replicates were randomly assigned to different analyses for the determination of the soil dissolved organic carbon (DOC) concentration (5/14), soil pH (2/14), microbial $CH_4$ respiration (6/14), and microbial community (1/14).

Microbial $CH_4$ respiration in anaerobic soils strongly depends on the production of soil DOC[3,45]. During the 160-day incubation period, the soil DOC concentrations under the three thermal treatments were periodically measured by destructively sampling the soil samples in the bottles. We found that the DOC concentration significantly increased (GKR: $P < 0.01$; TP: $P < 0.05$) with the increase in temperature and was not affected by the incubation duration (both wetland soils: $P > 0.05$) (Supplementary Fig. 5) across the whole incubation period. This result suggests that dissolved C substrate was more readily available at a higher than a lower temperature, in which case the substrate supply should have been sufficient for microbial methanogenesis in the naturally C-rich wetland soils during the incubation period. Furthermore, we found that the pH did not change significantly from the beginning to the end of the 160-day incubation period under any of the three thermal treatments ($P > 0.05$ for both wetland soils). Following the incubation period, soils from the three thermal treatments were used in short-term assays (Fig. 1b).

**Short-term assays**. We evaluated the compensatory response of soil microbial $CH_4$ respiration by conducting a short-term respiration assay[16,17,23,29]. The short

timescale of the assay prevented microbial adaptation to the assay temperatures. At the end of the main incubation, the soils from each 160-day thermal treatment were equally divided into 3 aliquots that were placed in thermostatic baths set at 8 °C, 12 °C, and 16 °C, and the $CH_4$ respiration rates were measured within 24 h (Fig. 1b). Headspace $CH_4$ and $CO_2$ concentrations and their [13]C contents were measured with cavity ring-down spectroscopy (Picarro G2201-i, USA) with a module for discrete small-volume gas samples (Picarro Small Sample Isotope Module, SSIM Ao314, USA). The amount of $CH_4$ in the aqueous phase was calculated using Henry's law after temperature correction[46]. The soil $CH_4$ respiration rate at each assay temperature (i.e., 8 °C, 12 °C, and 16 °C) was calculated on the basis of the soil mass, incubation time, gas accumulation, and headspace volume[44].

In most environments, $CH_4$ is mainly produced from either acetate or $CO_2$ and $H_2$. The relative importance of these two methanogenic pathways to total $CH_4$ production can be indicated by the C isotopic fractionation factor ($\alpha_{app}$)[47]. We calculated $\alpha_{app}$ using the $\delta^{13}$C values of the gas products $CH_4$ and $CO_2$, demonstrating that acetate-dependent methanogenesis was the dominant methanogenic pathway in the soils from the studied wetlands (Supplementary Fig. 6).

Acetate is one of the simplest forms of dissolved organic C available in anaerobic soils and is directly used by methanogens[48]. The addition of acetate can allow more complex fermentation and decomposition processes to be bypassed and ensures sufficient substrate availability[3,48]. Thus, acetate solution was added to the soils at a dose of 5 mg C L$^{-1}$ soil slurry to prevent the confounding effect of substrate availability when assessing the response of methanogenesis to temperature change in the short-term assay. Since acetate addition may affect methanogens and soil conditions, dose-response experiments were performed to confirm that the dose was sufficient but not excessive (Supplementary Fig. 7). In addition, the soil pH was adjusted to minimize the changes in soil pH caused by acetate addition[49].

Instantaneous temperature compensation does not affect $R_{mass}$[50]. To assess the compensatory thermal response of soil microbial $CH_4$ respiration at the different assay temperatures, it was necessary to control for methanogenic microbial biomass, as biomass regulates soil respiration rates[51]. The compensatory response was thus tested on a mass-specific basis[16,17,34]. To do so, $CH_4$-$R_{mass}$ was calculated by dividing the $CH_4$ respiration rates by the methanogenic microbial biomass. We used the abundance of methanogenic mcrA gene copies[52–54] to estimate the biomass of methanogenic microbes, as described below.

**Molecular analysis of microbial community composition and biomass**. RNA-based approaches are useful for the assessment of soil biogeochemical processes driven by active microbes[55–57]. Microbial RNA was extracted from 2-ml soil slurry samples at the end of the 160-day incubation period using the RNeasy PowerSoil Total RNA Kit according to the manufacturer's instructions (Qiagen, Germany). The final RNA pellet was suspended in 25 μL of RNase/DNase-free water. The RNA was reverse-transcribed into cDNA using PrimeScript RT Reagent Kits (TaKaRa, Japan). The RNA and cDNA concentrations were measured with a Qubit 3.0 fluorometer (Thermo Fisher Scientific, USA), and the cDNA samples were stored at −80°C until analysis.

Amplicon libraries of the mcrA gene[54] were prepared and loaded on an Illumina MiSeq instrument (Illumina, USA) to generate 2×300 paired-end reads. Raw sequencing reads were processed with VSEARCH v.1.9.6[58] and QIIME v.1.9.1 software pipelines[59]. In QIIME, paired-end sequences were concatenated into a single sequence and then filtered to remove sequences of low quality, sequences shorter than 200 bp and ambiguous nucleotides. Sequences were checked for chimeras and then clustered into OTUs using the de novo method at an 89% sequence similarity cutoff[54]. After quality filtering and rarefaction of the number of sequences in each sample to 22,522, 495,484 sequences remained and were included in all subsequent analyses. A representative sequence from each OTU was searched against the GenBank repository using the BLAST function. The weighted UniFrac value was calculated to assess β-diversity[60].

SYBR Green I-based qPCR of the methanogenic functional (mcrA) gene[54] was conducted using cDNA. Reactions were performed in triplicate for each sample. Known copy numbers of linearized plasmid DNA with the mcrA gene inserted from pure clones were used as standards for the quantifications. qPCR was conducted using a LightCycler 96 Instrument (Roche Molecular Systems, USA). Gene abundance was expressed as copies g$^{-1}$ soil dry weight. The abundance of mcrA gene copies was used to estimate the biomass of active methanogens.

**Calculation of the magnitude of compensatory responses and $Q_{10}$**. To assess how well the $R_{mass}$ of methanogens compensated for a change in temperature, we calculated the MCR of microbial $CH_4$ respiration under experimental warming or cooling (see Supplementary Fig. 8 for a theoretical representation and calculation details)[61,62]. The temperature sensitivity of the soil $CH_4$ respiration rate was expressed as a $Q_{10}$ value[63], which represents the factor by which respiration changes with every 10 °C increase in temperature. We used an exponential model[44,64,65] (Eq. [1]) to describe the relationship between $CH_4$-$R_{mass}$ and the assay temperature:

$$y = ae^{bT} \qquad (1)$$

where $y$ is the $CH_4$-$R_{mass}$ at assay temperature $T$ (°C), and $a$ and $b$ are fitted parameters. The $Q_{10}$ values were then calculated as follows:

$$Q_{10} = e^{10b} \qquad (2)$$

**Statistical analysis**. A linear mixed-effects model was used to test for incubation and assay temperature effects on soil $CH_4$ respiration and $CH_4$-$R_{mass}$. The fixed factors were the incubation temperature (RT − 4 °C, RT, and RT + 4 °C) and assay temperature (8, 12, and 16 °C), and the random factor was the spatial replicates within each wetland. Given that $CH_4$-$R_{mass}$ is essentially a ratio, we used the above model structure and a covariate approach to evaluate the treatment effects[18]. The covariate approach involved the inclusion of methanogenic biomass as the covariate and $CH_4$ respiration as the dependent variable.

The methanogen biomass, $Q_{10}$ values, and proportions of dominant methanogenic archaeal families in each wetland were treated statistically by one-way ANOVA, with the three thermal treatments during the main incubation period as the fixed factor. Pairwise multiple comparisons among the three thermal treatments were conducted using the Tukey HSD test at $P < 0.05$. Using a general linear model, repeated-measures ANOVA was implemented to test for differences in DOC concentration attributable to incubation time (days), thermal treatment, and their interaction. The linear relationship between the MCR and the β-diversity of the methanogenic community was tested by a linear mixed-effects model with temperature treatment (i.e., warming or cooling by 4 °C) as a random effect. Data that did not meet the assumptions of normality and homogeneity of variance were log-transformed before statistical testing. The linear mixed-effects model analysis was performed with the lme4 package in R (version 3.4.2), and $R^2$ and $P$ values were calculated with the MuMln and lmerTest packages, respectively.

**Reporting summary**. Further information on research design is available in the Nature Research Reporting Summary linked to this article.

## Data availability

Sequence data generated in the present study were deposited in NABI GenBank Short Read Archive (SRA) under accession number PRJNA668471 and the National Omics Data Encyclopedia (NODE, http://www.biosino.org/node) under accession number OEP000738 (Project ID). All data for this paper will be publicly available at: https://doi.org/10.5281/zenodo.4082274.

## Code availability

The code used in this study is available from the corresponding author upon request.

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

## Acknowledgements

We thank Jinsheng He, Zhaolei Li, Chenhao Zhou and Jia Yao for help in the field sampling and laboratory analyses. This work was supported by the National Key Research and Development Program of China (2018YFC1406402) and the National Science Foundation of China (91951112, 32030067 and 31830009).

## Author contributions

M.N. developed the original ideas presented in the manuscript; H.C. designed the research with the assistance from M.N., H.C. and T.Z. conducted the overall experiment and measurements with the assistance from lab assistants; H.C. and T.Z. analyzed the data with the assistance from M.N., B.L., and C.F.; H.C. and M.N. wrote the first draft, and all authors jointly revised the manuscript.

## Competing interests

The authors declare no competing interests.
