## [Peer Review File · Nature Communications]

REVIEWER COMMENTS

Reviewer #1 (Remarks to the Author):

The manuscript by Chen et al present data from a series of experiments investigating thermal acclimation of CH₄ production in peatland soils. Thermal acclimation has been extensively investigated for soil respiration but much less work has been done for CH₄ flux so this paper is timely and novel.

I have comments that the authors may wish to consider in their revision of the manuscript.

Acclimation – The manuscript is built around a narrative on thermal acclimation. In my view what the authors are measuring here is not thermal acclimation. Thermal acclimation is defined as physiological adjustments that cells (or individuals) make to respond to changes in the environment, usually in order to maintain cellular homeostasis. Thermal acclimation is a term used extensively in the plant physiology literature where individual plants are transplanted into different environments and physiological traits are measured within generations. This is opposed to thermal adaptation which accounts for genetic evolution – i.e. changes in genotypes in response to changes in temperature. The experiment in the present manuscript involves incubating complex communities of microbes under different temperature conditions and as such a multitude of processes will be contributing to the emergent outcomes they observe. Sure there will be some physiological acclimation in the very early stages of the experiment, however the bulk of what they observe will be driven by (a) turnover in the taxonomic composition of the community – i.e. ecological selection of taxa better suited to the prevailing environment, and (b) evolutionary adaptation of those taxa that are present over the duration of the incubation – i.e. evolutionary selection for genotypes within populations that are better suited to the novel environment. One aspect of this paper that I think is particularly strong is the fact that the authors are actually quantifying the taxonomic composition and how it changes. Thus they are well aware that a large part of what they are observing is related to taxonomic turnover.

I suggest that the authors devote some careful thought to the above and spend some time in the manuscript delving a bit more deeply into the mechanism that could be driving the outcomes they observe. The use of the term acclimation is wrong here in my view for the reasons articulated above. I understand why the authors have done this and the reason lies in the mess and confusion over term and mechanisms that is prevalent in the microbial soil respiration literature. That said, the authors here have an opportunity to lay this out in the introduction, clearly defining acclimation, adaptation and species turnover as contributing factors and then investigating (at least speculatively) the extent to which their findings are related to this 3 mechanisms. I suspect that the majority of their result is driven by turnover as evidenced by their very nice Fig. 4. What I don't understand with Fig 4 is why they present the x axis as % change? For Fig. 4b they show % change in alpha diversity relative to the control, it is notable that all values are > 0. Does this mean that species richness increased in both the warmed and the cooled environments? Seems unlikely. Why don't the authors just express MAT as a function of the alpha diversity of the initial community prior to the warming/cooling incubation? That way they can actually test some hypotheses about what might be driving the outcomes. For example, high diversity communities would be expected to have greater thermal trait diversity upon which to select on (across species) and therefore if species turnover is a dominant mechanism then they should see a positive association between initial alpha diversity and MAT. Similarly, the % change in beta diversity is even more confusing. Beta diversity is a measurement of taxonomic turnover between two communities. Thus, the beta diversity between control vs treatment incubations tells you the magnitude of species replacements, there is no need to then express this as a % change.

To biomass correct or to not biomass correct? – The authors analyses of the temperature response are all expressed as mass specific rates of CH₄ production. The argument here is that to reveal 'acclimation' differences between bottles in terms of biomass need to be accounted for. I get that. However again I think the authors are missing a trick here. Ultimately what we care about from a

climate change perspective is how will temperature affect the amount of CH₄ produced over the long term. As the authors point out, a large part of the driving force behind this response will be due to changes in microbial biomass that in turn affect the amount of CH₄ produced. So to get at the core of what is driving the long term temperature response, we want to understand what is happening in terms of the biomass response as this will be a major contributor to the emergent outcome in the real world. Why don't the authors analyse the response of total CH₄ production and then use their biomass data as a variable to explain how and why total CH₄ flux is being modulated over the long term. The changes in biomass in response to warming/cooling are part of the community response and are just as important as species turnover. At present the authors are not making the most of their rich dataset in interrogating the possible mechanisms that give rise to the long term temperature response.

Reviewer #2 (Remarks to the Author):

Chen et al present an interesting and novel study on the acclimation of methanogenic microbial consortia to long-term temperature change and whether this acclimation dampens the production of methane under warming. Overall, I found the study to be technically sound and convincing – I did not have any specific comments on the methods. In places, the data analysis was not clear, but I suspect this is because the manuscript has been written in a condensed format. With a more expansive description of the methods and data analysis (including justification and relevance) early in the manuscript itself I believe this study would be stronger and more convincing. The formatting at Nature Communications would allow this, I believe. Below I highlight some more specific comments on where I think some further discussion/justification is warranted. But overall, the manuscript could do with more information as it took a while to decipher what was done and why, especially early on from lines 69-87 (figure included). More transparency in data treatment would also be welcome. Otherwise this is a well written and well-presented study.

L56-60. This sentence does not make sense to me as a justification for why the study regions represent good model systems for global patterns.

L89. It should be more clearly stated that the observed results are only from the 24-hour assay at the end of the two different incubation periods ("spin-up" and then the different treatments). Right now, this descriptive section on the methods is difficult to follow what is actually measured.

L101. So with a high pH these soils do not represent a good model system for high altitude peatlands in general?

Fig. 2c,d. Why have these ratios been natural log transformed?

Fig. 4. These R²-values look to be strengthened by combining both high altitude and high latitude peatlands responses. What is the justification for combining both peatlands in this analysis when previously (extended data fig. 1) the magnitude of response by the two peatland types was different? What do the diversity index R²-values look like for the individual peatland types alone?

L176-181. The results suggest that the observed methanogenic response is linked only to change in the microbial population, not the rise and/or fall of specific microbial families as there appears to be no consistent response between the two types of peatlands (again a bit confusing as to whether the peatland comparisons should be combined or examined separately – see previous comment). The conclusion seems to be that shifts in microbial composition may be important, but this does not rule out functional redundancy because it is not clear which specific taxa are doing what (the level of data analysis does not currently allow for this extended conclusion). It does suggest the response is different to microbial CO₂ respiration, which is an interesting finding. The following paragraph puts this into context, and this finding is not oversold in the abstract (L24-25).

L190-193. This is a big "if". The manuscript could do more early on to make it clear that the studied systems can in fact represent wider peatland soil patterns (see comment on L56-60).

L188-201 including final conclusions. This last paragraph is very short and lacks context. Would be nice to see some data from other studies and further comparison done in a more fleshed out discussion. Right now, the study lacks context and proper comparisons and the data largely stands alone, making it harder to perceive the novelty of the findings.

Reviewer #3 (Remarks to the Author):

General comments

This study investigates the thermal acclimation of methanogenesis in incubations of soils collected from high latitude and altitude peatlands in China. The study addresses an important knowledge gap. Although it is well known that rates of methanogenesis generally increase with temperature or warming, the magnitude of the temperature response varies widely in previous studies. As the authors point out in their introduction, few to no studies have examined the acclimation of methanogens to temperature, especially over the longterm and in indigenous microbial communities from field samples. Peatlands comprise an important subset of freshwater wetlands that predominate over methane emission on a global scale. Since peatlands store such a large portion of terrestrial soil carbon, it is very important to understand the response of methanogenesis to warming in peat soils in order to understand climate feedbacks and build better climate models.

I find the study intriguing and interesting. The experimental design of the study is straightforward and employs state-of-the-field methods. The investigation is efficiently executed and the paper is well-written. I find that the rationale of thermal acclimation, as shown in Fig. 1, is clear and carefully explained. The conclusions of the study are that methanogens in peatland soils acclimate to temperature and that microbial community dynamics play a vital role in thermal acclimation. The results support a significant, albeit incremental, change in mass-specific rates of methanogenesis with warming or cooling treatment. Microbial community composition and diversity also appear to show differences between treatments. Unfortunately, there are a number of drawbacks to the study. The soils and incubation conditions are not representative of global peatlands that contain the most terrestrial soil carbon on a global scale. Results from microbial community analysis are contradictory and do not strongly support conclusions from rate measurements. The sequence dataset used to investigate microbial community dynamics is limited. The abovementioned drawbacks to the study take away from its potential significance to peatlands over broad scales.

Specific comments

Extensive statistical comparisons indicate that mass-specific rates of methanogenesis differ between temperature treatments. However, these differences are small. As shown in extended data Fig. 1, the magnitude of thermal acclimation ranges from 1 to 3. Microbial communities vary over exponential scales in the laboratory as well as in the environment. Even in pure culture, the significance of a change in growth or biomass by a factor of 1 to 3 is questionable.

The study sites and incubation conditions do not appear to be representative of globally significant, climatically sensitive peatlands. The majority of global peatlands which predominate over greenhouse gas emission are acidic, nutrient-poor, and devoid of inorganic terminal electron acceptors (TEAPs). This study employs soils that are mildly acidic to basic and contain substantial concentrations of TEAPs. With a soil pH range of 5.5 to 8, these soils are on the extreme high end of the pH range for peatlands. Iron(III) content, shown in extended data Fig. 2, is also at the high end for most peatland soils. Moreover, rates determined for thermal acclimation were measured in 24 hour incubations amended with acetate. Amendment with acetate indicates that thermal acclimation was studied under

enrichment culture conditions which are unlike the in situ condition. Acetate can effect microbial communities in many ways. Besides serving as a substrate for methanogenesis, acetate has been observed to accumulate in peat and inhibit methanogenesis. In soils with pH lower than the pKa of acetic acid (4.8), this acid often accumulates and can be toxic to methanogens. Thus, the applicability of these results to global peatlands can be called into question.

More importantly, the authors achieve experimental warming with an incubation of 160 days. They interpret this incubation as "long-term" experimental warming. This is not long-term. In contrast to aerobic respiration in upland dryer soils, acclimation of anaerobic respiration in freshwater wetlands is usually only observed in field warming experiments after years of treatment and incubation studies are conducted over much longer time scales. For example, see studies by Knoblauch et al 2018 *Nature Climate Change* and Schadel et al. 2014 *Global Change Biology*, which incubated peat soils for 1 to 7 years. I question whether the activity and biomass determined under these conditions is representative of thermal acclimation that may or may not occur in situ. In field warming experiments conducted recently in peatlands (Hopple et al., 2020. *Nature Communications*, 11(1), pp.1-7), the biomass and composition of methanogen communities were observed to be stable, even after 4 to 5 years of warming. Methanogen communities in soils often show a large natural heterogeneity in biomass and community composition, even at the same site. Biomass is determined using qPCR as a proxy and the qPCR data is not provided in the study. Statistical analysis is performed on normalized rates. Thus it is not clear whether methanogen biomass changes with temperature treatment. The authors should provide the data that were used to generate Fig. 1 in the supplement. Was activity, biomass, or both different between temperature treatments?

I find the results on microbial community dynamics to be equivocal and decoupled from the rate measurements. Isotopic fractionation (extended data Fig. 4) indicates that acetoclastic methanogenesis is dominant at incubated soils. However, contrary to the mass-specific rate measurements, acetoclastic methanogens (*Methanotrichaceae*) actually decline in relative abundance with warming or cooling treatment in the high latitude soil incubations (Fig. 3). An increase in the relative abundance of acetoclastic methanogens with warming treatment is only observed in the high altitude peatland soils where the magnitude of thermal acclimation is much lower, around a factor of 1.

The interpretation of trends in diversity and community composition is misleading. In Fig 4., the relationship between MTA and diversity appears to be largely driven by values from the high latitude incubations, which do not show strong trends in methanogen relative abundance with temperature treatment. Trends in diversity should be statistically analyzed separately for each incubation, HLP and HAP sites. The authors conclude that "changes in the methanogenic community structure might be important in regulating the thermal acclimation of methanogenesis." If this is the case, then why was this trend not reflected in the relative abundance of methanogens in Fig. 3b?

Responses to Reviewer #1

[Comment 1] The manuscript by Chen et al present data from a series of experiments investigating thermal acclimation of CH₄ production in peatland soils. Thermal acclimation has been extensively investigated for soil respiration but much less work has been done for CH₄ flux so this paper is timely and novel.

[Response] Thanks for your positive assessments. Also, thanks to your insightful comments, which have significantly improved the quality of this work.

[Comment 2] Acclimation – The manuscript is built around a narrative on thermal acclimation. In my view what the authors are measuring here is not thermal acclimation. Thermal acclimation is defined as physiological adjustments that cells (or individuals) make to respond to changes in the environment, usually in order to maintain cellular homeostasis. Thermal acclimation is a term used extensively in the plant physiology literature where individual plants are transplanted into different environments and physiological traits are measured within generations. This is opposed to thermal adaptation which accounts for genetic evolution – i.e. changes in genotypes in response to changes in temperature. The experiment in the present manuscript involves incubating complex communities of microbes under different temperature conditions and as such a multitude of processes will be contributing to the emergent outcomes they observe. Sure there will be some physiological acclimation in the very early stages of the experiment, however the bulk of what they observe will be driven by (a) turnover in the taxonomic composition of the community – i.e. ecological selection of taxa better suited to the prevailing environment, and (b) evolutionary adaptation of those taxa that are present over the duration of the incubation – i.e. evolutionary selection for genotypes within populations that are better suited to the novel environment. One aspect of this paper that I think is particularly strong is the fact that the authors are actually quantifying the taxonomic composition and how it changes. Thus they are well aware that a large part of what they are observing is related to taxonomic turnover. I suggest that the authors devote some careful thought to the above and spend some time in the manuscript delving a bit more deeply into the mechanism that could be driving the outcomes they observe. The use of the term acclimation is wrong here in my view for the reasons articulated above. I understand why the authors have done this and the reason lies in the mess and confusion over term and mechanisms that is prevalent in the microbial soil respiration literature. That said, the authors here have an opportunity to lay this out in the introduction, clearly defining acclimation, adaptation and species turnover as contributing factors and then investigating (at least speculatively) the extent to which their findings are related to this 3 mechanisms. I suspect that the majority of their result is driven by turnover as evidenced by their very nice Fig. 4.

[Response] Thanks for the insightful comments and helpful suggestions. We used a more accurate term ‘compensatory response’ to reflect what we observed throughout the revised

version (please see e.g., page 2, lines 14-15). In the *Introduction* section of the revised manuscript, we have clarified the potential mechanisms (i.e. acclimation, adaptation and species turnover) of microbial compensation response to eliminate the confusion of terms: “*These adjustments in soil microbial respiration rate responses to temperature changes could result from acclimation (physiological responses of individuals), adaptation (genetic variation within species) and/or species turnover (shifts in the species composition of the community)*^{15,19-21}.” (please see pages 2-3, lines 42-45).

In addition, we have further highlighted that the shifts in the community composition of methanogens play a vital role in regulating the compensatory response of methanogenesis to temperature change in the *Results and discussion* section of the revision: e.g., “*Changes in community structure are often considered the key mechanism by which plant communities maintain their functioning under the changing external environment*^{37,38}. *Likewise, we observed that the shifts in community composition of methanogens were positively associated with the magnitude of their compensatory thermal responses (Fig. 4), reducing the extent to which CH₄ respiration rates respond to temperature change.*”, “*However, our findings suggest that changes in the methanogenic community structure might be responsible for the compensatory responses of microbial methanogenesis to temperature change, being inconsistent with the role of community dynamics in the response of microbial CO₂ respiration*⁴¹.” (please see page 8, lines 173-177 and 179-182).

[Comment 3] For Fig. 4b they show % change in alpha diversity relative to the control, it is notable that all values are > 0. Does this mean that species richness increased in both the warmed and the cooled environments? Seems unlikely.

[Response] In the original Fig. 4b, the large scale (ranging from -100 to 400) of X-axis caused the values of the relative change in Shannon index from HAP sites to be concentrated, which made them visually appear to be greater than 0, but actually not all of the values > 0, with values ranging from -12.63 to 1.54.

[Comment 4] What I don't understand with Fig 4 is why they present the x axis as % change? Why don't the authors just express MTA as a function of the alpha diversity of the initial community prior to the warming/cooling incubation? That way they can actually test some hypotheses about what might be driving the outcomes. For example, high diversity communities would be expected to have greater thermal trait diversity upon which to select on (across species) and therefore if species turnover is a dominant mechanism then they should see a positive association between initial alpha diversity and MTA. Similarly, the % change in beta diversity is even more confusing. Beta diversity is a measurement of taxonomic turnover between two communities. Thus, the beta diversity between control vs treatment incubations tells you the magnitude of species replacements, there is no need to then express this as a % change.

[Response] Thanks for the comments. According to the 2nd Comment, we replaced ‘the magnitude of thermal acclimation (MTA)’ with ‘the magnitude of compensatory response (MCR)’ in the revised manuscript. By definition (Loveys *et al.*, 2003; Reich *et al.*, 2016), MCR reflects the proportion of the initial rise/decline that is recovered following compensation [please see Fig. R1, $MCR_{warming} = (A-C)/(A-B)$; $MCR_{cooling} = (A'-C)/(A'-B)$]. Since MCR is essentially a ratio of relative changes, we expressed MCR as a function of the % change in α -diversity. Indeed, we attempted to establish relationships between MCR and the α -diversity of the initial methanogenic community prior to the warming/cooling incubation, but the results were not statistically significant ($P > 0.05$).

Furthermore, the β -diversity was not expressed as % change in the previous version. To avoid confusion arising from the expression as ‘relative change’, we have focused on the β -diversity to show the magnitude of species turnover between the control (RT) and the treatment in the revision. In addition, according to the other two reviewers’ comments and suggestions (“*These R²-values look to be strengthened by combining both high altitude and high latitude peatlands responses. What do the diversity index R²-values look like for the individual peatland types alone?*” “*Trends in diversity should be statistically analysed separately for each incubation, HLP and HAP sites.*”), the two types of soils were analyzed separately in the revision (please see Fig. 4). Hope these adjustments would make the results clearer.

Fig. R1 Theoretical temperature response curves of soil mass-specific respiration (R_{mass}) under warming and cooling. The reference temperature (RT) was set at 12°C. Under experimental warming, the magnitude of compensatory responses (MCR) of methanogenesis is a measure of how much of the respiratory increase (e.g., A minus B) expected due to short-term warming (minutes to hours) is eliminated by physiological compensation (e.g., A minus C) of R_{mass} under warming to the same extent but over a period of months. Thus, the

MCR under experimental warming was calculated as follows: $MCR_{\text{warming}} = (A-C)/(A-B)$. As compensatory community responses, they must show opposite effects of warming and cooling, the MCR under experimental cooling was thus calculated as follows: $MCR_{\text{cooling}} = (A'-C')/(A'-B)$.

[Comment 5] To biomass correct or to not biomass correct? – The authors analyses of the temperature response are all expressed as mass specific rates of CH₄ production. The argument here is that to reveal ‘acclimation’ differences between bottles in terms of biomass need to be accounted for. I get that. However again I think the authors are missing a trick here. Ultimately what we care about from a climate change perspective is how will temperature affect the amount of CH₄ produced over the long term. As the authors point out, a large part of the driving force behind this response will be due to changes in microbial biomass that in turn affect the amount of CH₄ produced. So to get at the core of what is driving the long term temperature response, we want to understand what is happening in terms of the biomass response as this will be a major contributor to the emergent outcome in the real world. Why don’t the authors analyse the response of total CH₄ production and then use their biomass data as a variable to explain how and why total CH₄ flux is being modulated over the long term. The changes in biomass in response to warming/cooling are part of the community response and are just as important as species turnover. At present the authors are not making the most of their rich dataset in interrogating the possible mechanisms that give rise to the long term temperature response.

[Response] In the revised manuscript, we have reanalyzed our data according to your insightful suggestions. Results show that the thermal response of microbial methanogenesis showed similar patterns regardless of whether soil CH₄ respiration was corrected or not corrected for biomass (please see pages 4-5, lines 93-101; Fig. 2 and Supplementary Fig. 1). Furthermore, given that the methanogen biomass did not differ significantly ($P > 0.05$) between control and treatment soils (Supplementary Fig. 2), the driving force behind this compensatory thermal response may not be the change in microbial biomass (please see pages 4-5, lines 98-101).

In addition, we have used the linear mixed-effects model and a covariate approach to evaluate the thermal treatment effects in the revised manuscript (please see Fig. 2). The covariate approach involved the inclusion of methanogen biomass as the covariate and CH₄ respiration as the dependent variable (please see page 13, lines 320-322).

Responses to Reviewer #2

[Comment 1] Chen et al present an interesting and novel study on the acclimation of methanogenic microbial consortia to long-term temperature change and whether this acclimation dampens the production of methane under warming. Overall, I found the study to be technically sound and convincing – I did not have any specific comments on the methods. In places, the data analysis was not clear, but I suspect this is because the manuscript has been written in a condensed format. With a more expansive description of the methods and data analysis (including justification and relevance) early in the manuscript itself I believe this study would be stronger and more convincing. The formatting at Nature Communications would allow this, I believe. Below I highlight some more specific comments on where I think some further discussion/justification is warranted. But overall, the manuscript could do with more information as it took a while to decipher what was done and why, especially early on from lines 69-87 (figure included). More transparency in data treatment would also be welcome. Otherwise this is a well written and well-presented study.

[Response] Thanks for the positive comments. These comments, together with those listed below helped us to further strengthen the methods and data analyses in the revised manuscript.

[Comment 2] L56-60. This sentence does not make sense to me as a justification for why the study regions represent good model systems for global patterns.

[Response] Following the suggestion of the *Reviewer#3* (“*The study sites do not appear to be representative of globally significant, climatically sensitive peatlands.*”), we have deleted the term ‘model’ and used a more general term ‘wetland’ instead of ‘peatland’ in the revised manuscript to eliminate this confusion. Besides, we have further clarified the use of the two wetland soils in the revised *Introduction* section:

“To discern whether microbial methanogenesis in anaerobic soils exhibits a compensatory response to temperature change, we collected wetland soil samples from plots established at four sites in the Greater Khingan Range (GKR) and four sites on the Tibetan Plateau (TP) (Supplementary Table 1). Because of the differences in the soil methanogenic community and physicochemical properties between the selected GKR and TP soils^{25,26}, there might be considerable differences in the thermal responses of methanogens to temperature change in these contrasting soils, and the use of these soils may convincingly test the compensatory response of microbial CH₄ respiration to changing temperature and its underlying mechanisms.” (please see page 3, lines 52-59).

[Comment 3] L89. It should be more clearly stated that the observed results are only from the 24-hour assay at the end of the two different incubation periods (“spin-up” and then the different treatments). Right now, this descriptive section on the methods is difficult to follow what is actually measured.

[Response] Thanks for the useful suggestion. In the revised manuscript, we have made it clear that the observed results were from the short-term assay by adding the text below: “We conducted short-term assays (Fig. 1b; see Methods).....” (please see page 4, line 88). Meanwhile, we have also highlighted what was measured for the short-term assays by adding a separate subheading in the relevant *Methods* section (please see page 11, line 249).

[Comment 4] L101. So with a high pH these soils do not represent a good model system for high altitude peatlands in general?

[Response] As mentioned above, in the revised manuscript, we have deleted the term ‘model’ and used a more general term ‘wetland’ instead of ‘peatland’.

[Comment 5] Fig. 2c,d. Why have these ratios been natural log transformed?

[Response] Sorry for the confusion. In the previous manuscript, for clarity we showed the log response ratios of the treatment effects for the variable $\text{CH}_4\text{-}R_{\text{mass}}$. That is, ratios > 0 indicate that $\text{CH}_4\text{-}R_{\text{mass}}$ rates were greater in the treatment than the control and vice-versa for ratios < 0 . We have explained this in this version (please see page 5, lines 108-110).

[Comment 6] Fig. 4. These R^2 -values look to be strengthened by combining both high altitude and high latitude peatlands responses. What is the justification for combining both peatlands in this analysis when previously (extended data fig. 1) the magnitude of response by the two peatland types was different? What do the diversity index R^2 -values look like for the individual peatland types alone?

[Response] Following this comment and the suggestion of the *Reviewer#3* (“Trends in diversity should be statistically analyzed separately for each incubation, HLP and HAP sites.”), two types of soils were analyzed separately to make our results more convincing (please see Fig. 4).

Furthermore, according to the suggestions of *Reviewer #1*, we have focused on the β -diversity to show the magnitude of species turnover between the control and the treatment in the revised manuscript, and found that the magnitude of thermal responses was positively related to the β -diversity of the methanogenic community under experimental warming and cooling for both GKR ($R^2 = 0.53$, $P = 0.014$) and TP ($R^2 = 0.63$, $P < 0.01$) soils (please see pages 7-8, lines 164-166; Fig.4).

[Comment 7] L176-181. The results suggest that the observed methanogenic response is linked only to change in the microbial population, not the rise and/or fall of specific microbial families as there appears to be no consistent response between the two types of peatlands (again a bit confusing as to whether the peatland comparisons should be combined or examined separately – see previous comment). The conclusion seems to be that shifts in microbial composition may be important, but this does not rule out functional redundancy because it is

not clear which specific taxa are doing what (the level of data analysis does not currently allow for this extended conclusion). It does suggest the response is different to microbial CO₂ respiration, which is an interesting finding. The following paragraph puts this into context, and this finding is not oversold in the abstract (L24-25).

[Response] Thanks for the useful suggestion. As mentioned above, we have reanalyzed the relationship between the magnitude of thermal response and diversity for each wetland soil (please see pages 7-8, lines 164-166; Fig.4).

Furthermore, following the reviewer's suggestion, we have deleted the discussion on the functional redundancy, and highlighted the differences in the role of community shifts in regulating microbial methanogenesis and microbial CO₂ respiration (please see page 8, lines 179-182). In the revised *Abstract* section, we have also emphasized the key importance of methanogenic community dynamics in regulating the response of soil CH₄ respiration to temperature change: "*Changes in the species composition of methanogenic community under warming and cooling largely explained the compensatory response in the soils.*" (please see page 2, lines 24-26).

[Comment 8] L190-193. This is a big "if". The manuscript could do more early on to make it clear that the studied systems can in fact represent wider peatland soil patterns (see comment on L56-60).

[Response] In order to properly interpret our research implications, and to avoid unduly extending the applicability of our results to broad scales, we have deleted the previous text of "*If such thermal acclimation is a common phenomenon in similar ecosystems around the world*" and "*If our results are broadly indicative of what will occur in wetlands around the world*" in the revised manuscript.

[Comment 9] L188-201 including final conclusions. This last paragraph is very short and lacks context. Would be nice to see some data from other studies and further comparison done in a more fleshed out discussion. Right now, the study lacks context and proper comparisons and the data largely stands alone, making it harder to perceive the novelty of the findings.

[Response] Thanks for the useful suggestion. We have strengthened the discussion and conclusion in the revised manuscript. Amendments in these sections are highlighted in red (e.g., please see pages 4-5, lines 93-101; pages 6-7, lines 141-151; page 8, lines 173-177 and 179-182; page 9, lines 188-195).

Responses to Reviewer #3

[Comment 1] This study investigates the thermal acclimation of methanogenesis in incubations of soils collected from high latitude and altitude peatlands in China. The study addresses an important knowledge gap. Although it is well known that rates of methanogenesis generally increase with temperature or warming, the magnitude of the temperature response varies widely in previous studies. As the authors point out in their introduction, few to no studies have examined the acclimation of methanogens to temperature, especially over the long term and in indigenous microbial communities from field samples. Peatlands comprise an important subset of freshwater wetlands that predominate over methane emission on a global scale. Since peatlands store such a large portion of terrestrial soil carbon, it is very important to understand the response of methanogenesis to warming in peat soils in order to understand climate feedbacks and build better climate models.

[Response] Thanks for the encouragements.

[Comment 2] I find the study intriguing and interesting. The experimental design of the study is straightforward and employs state-of-the-field methods. The investigation is efficiently executed and the paper is well-written. I find that the rationale of thermal acclimation, as shown in Fig. 1, is clear and carefully explained. The conclusions of the study are that methanogens in peatland soils acclimate to temperature and that microbial community dynamics play a vital role in thermal acclimation. The results support a significant, albeit incremental, change in mass-specific rates of methanogenesis with warming or cooling treatment. Microbial community composition and diversity also appear to show differences between treatments. Unfortunately, there are a number of drawbacks to the study. The soils and incubation conditions are not representative of global peatlands that contain the most terrestrial soil carbon on a global scale. Results from microbial community analysis are contradictory and do not strongly support conclusions from rate measurements. The sequence dataset used to investigate microbial community dynamics is limited. The abovementioned drawbacks to the study take away from its potential significance to peatlands over broad scales.

[Response] These comments, together with those listed below have greatly improved our manuscript. Thank you!

[Comment 3] Extensive statistical comparisons indicate that mass-specific rates of methanogenesis differ between temperature treatments. However, these differences are small. As shown in extended data Fig. 1, the magnitude of thermal acclimation ranges from 1 to 3. Microbial communities vary over exponential scales in the laboratory as well as in the environment. Even in pure culture, the significance of a change in growth or biomass by a factor of 1 to 3 is questionable.

[Response] Sorry for the confusion. According to the suggestions of *Reviewer #1*, we have

replaced ‘the magnitude of thermal acclimation (MTA)’ with ‘the magnitude of compensatory response (MCR)’ in the revised manuscript, to avoid the confusion caused by the term “acclimation”.

By definition, MCR is a measurement of how well microbial methanogenesis compensate for an initial change in temperature, which reflects the proportion of the initial rise/decline that is recovered following compensation (please see Fig. R1). For instance, as shown in Fig. R1, the MCR under experimental warming was calculated as follows: $MCR_{\text{warming}} = (A-C)/(A-B)$, where $[A-B]$ represents the respiratory increase expected due to short-term warming (minutes to hours), and $[A-C]$ represents the physiological compensation of R_{mass} under warming to the same extent but over a period of months. MCR is a ratio of relative change, which is not absolute changes in microbial growth or biomass. Currently, MCR has been widely used for assessing biotic compensatory responses (e.g., Atkin and Tjoelker, 2003; Loveys *et al.*, 2003; Reich *et al.*, 2016). We calculated MCR of Bradford *et al.* (2008) [Title: Thermal adaptation of soil microbial respiration to elevated temperature] using their data of Fig. S2-S4, and found that their MCR values range approximately from 0 to 6, covering our MCR values. Thanks for your understanding!

Fig. R1 Theoretical temperature response curves of soil mass-specific respiration (R_{mass}) under warming and cooling. The reference temperature (RT) was set at 12°C. Under experimental warming, the magnitude of compensatory responses (MCR) of methanogenesis is a measure of how much of the respiratory increase (e.g., A minus B) expected due to short-term warming (minutes to hours) is eliminated by physiological compensation (e.g., A minus C) of R_{mass} under warming to the same extent but over a period of months. Thus, the MCR under experimental warming was calculated as follows: $MCR_{\text{warming}} = (A-C)/(A-B)$. As compensatory community responses, they must show opposite effects of warming and cooling,

the MCR under experimental cooling was thus calculated as follows: $MCR_{cooling} = (A'-C)/(A'-B)$.

[Comment 4] The study sites and incubation conditions do not appear to be representative of globally significant, climatically sensitive peatlands. The majority of global peatlands which predominate over greenhouse gas emission are acidic, nutrient-poor, and devoid of inorganic terminal electron acceptors (TEAPs). This study employs soils that are mildly acidic to basic and contain substantial concentrations of TEAPs. With a soil pH range of 5.5 to 8, these soils are on the extreme high end of the pH range for peatlands. Iron(III) content, shown in extended data Fig. 2, is also at the high end for most peatland soils. Moreover, rates determined for thermal acclimation were measured in 24 hour incubations amended with acetate. Amendment with acetate indicates that thermal acclimation was studied under enrichment culture conditions which are unlike the in situ condition. Acetate can effect microbial communities in many ways. Besides serving as a substrate for methanogenesis, acetate has been observed to accumulate in peat and inhibit methanogenesis. In soils with pH lower than the pKa of acetic acid (4.8), this acid often accumulates and can be toxic to methanogens. Thus, the applicability of these results to global peatlands can be called into question.

[Response] Following this comment and comments from the *Reviewer#2*, we have used a more general term 'wetland' instead of 'peatland', and removed the sentences about peatlands from the original *Introduction* section. In addition, in order to properly interpret our research implications and to avoid over-generalization of results, we have deleted the previous text of "*If such thermal acclimation is a common phenomenon in similar ecosystems around the world*" and "*If our results are broadly indicative of what will occur in wetlands around the world*" in the revised manuscript.

For acetate, it is one of the simplest forms of dissolved organic C available in anaerobic soils and is directly used by methanogens. At the end of the 160-day incubation, our isotopic fractionation demonstrated that acetate-dependent methanogenesis was the dominant methanogenic pathway in the studied soils (please see Supplementary Fig. 6). Therefore, in this study, the addition of acetate can allow more complex fermentation and decomposition processes to be bypassed and also prevent the confounding effect of substrate availability when assessing the response of methanogens to temperature change. In order to minimize the potential adverse effects of acetate addition on methanogens and soil conditions, dose-response experiments were performed to confirm that the dose was sufficient but not excessive (please see Supplementary Fig. 7). In addition, the soil pH was adjusted to minimize the changes in soil pH caused by acetate addition (please see page 11, lines 269-272). Thanks for your consideration!

[Comment 5] More importantly, the authors achieve experimental warming with an incubation

of 160 days. They interpret this incubation as “long-term” experimental warming. This is not long-term. In contrast to aerobic respiration in upland dryer soils, acclimation of anaerobic respiration in freshwater wetlands is usually only observed in field warming experiments after years of treatment and incubation studies are conducted over much longer time scales. For example, see studies by Knoblauch et al 2018 *Nature Climate Change* and Schadel et al. 2014 *Global Change Biology*, which incubated peat soils for 1 to 7 years. I question whether the activity and biomass determined under these conditions is representative of thermal acclimation that may or may not occur in situ. In field warming experiments conducted recently in peatlands (Hopple et al., 2020. *Nature Communications*, 11(1), pp.1-7), the biomass and composition of methanogen communities were observed to be stable, even after 4 to 5 years of warming. Methanogen communities in soils often show a large natural heterogeneity in biomass and community composition, even at the same site. Biomass is determined using qPCR as a proxy and the qPCR data is not provided in the study. Statistical analysis is performed on normalized rates. Thus it is not clear whether methanogen biomass changes with temperature treatment. The authors should provide the data that were used to generate Fig. 1 in the supplement. Was activity, biomass, or both different between temperature treatments?

[Response] In the initial design of the experiment, we hypothesized that the 160 days is the time to verify whether there is thermal acclimation/adaptation with reference to previous incubation studies [e.g., Hartley *et al.*, *Ecology letters*, 2008 (200 days); Bradford *et al.*, *Global Biology Change*, 2010 (77 days); Karhu *et al.*, *Nature*, 2014 (90 days); Bradford *et al.*, *Nature Ecology & Evolution*, 2019 (100 days)]. As expected, we observed compensatory responses of microbial methanogenesis to ambient temperature after 160 days. However, the use of the term ‘long-term’ may be problematic according to your suggestion, and we have removed “long-term” from the manuscript. Anyway, this is a good way to avoid confusion caused by time scale.

Furthermore, the thermal response of microbial methanogenesis showed similar patterns regardless of whether soil CH₄ respiration was corrected or not corrected for biomass. We have added a new Supplementary Fig. 2 about biomass in this version. Given that the methanogen biomass did not differ significantly ($P > 0.05$) between control and treatment soils, the driving force behind this compensatory thermal response may not be the change in microbial biomass (please see pages 4-5, lines 93-101; Fig. 2 and Supplementary Fig. 1, 2).

In addition, we have used the linear mixed-effects model and a covariate approach to evaluate the thermal treatment effects in this version (please see Fig. 2). The covariate approach involved the inclusion of methanogen biomass as the covariate and CH₄ respiration as the dependent variable (please see page 13, lines 320-322). Thanks for your consideration!

[Comment 6] I find the results on microbial community dynamics to be equivocal and decoupled from the rate measurements. Isotopic fractionation (extended data Fig. 4) indicates that acetoclastic methanogenesis is dominant at incubated soils. However, contrary to the

mass-specific rate measurements, acetoclastic methanogens (Methanotrichaceae) actually decline in relative abundance with warming or cooling treatment in the high latitude soil incubations (Fig. 3). An increase in the relative abundance of acetoclastic methanogens with warming treatment is only observed in the high altitude peatland soils where the magnitude of thermal acclimation is much lower, around a factor of 1.

[Response] In the revised manuscript, we have strengthened the discussion of the inconsistent changes between microbial CH₄ respiration rates and the relative abundance of methanogens under warming/cooling:

“However, in contrast to the consistent changes in CH₄-R_{mass} for both soils (Fig. 2c,d), there was no consistent response to temperature change for the dominant methanogenic archaea between the two soil types. For instance, Methanotrichaceae in GKR soils significantly ($P < 0.05$) declined in relative abundance under both experimental warming and cooling, while it significantly increased in relative abundance with increasing temperature in TP soils (Fig. 3b). The relative abundance refers to the evenness of distribution of individuals among species in a community, and the observed differences in relative abundance may not accurately reflect the quantity of the microbial community and the intersample differences between taxa^{35,36}. These may be the reasons why the shifts in the relative abundance of specific methanogens could not explain the compensatory response of microbial methanogenesis to temperature change.” (please see pages 6-7, lines 141-151).

[Comment 7] The interpretation of trends in diversity and community composition is misleading. In Fig 4., the relationship between MTA and diversity appears to be largely driven by values from the high latitude incubations, which do not show strong trends in methanogen relative abundance with temperature treatment. Trends in diversity should be statistically analyzed separately for each incubation, HLP and HAP sites. The authors conclude that “changes in the methanogenic community structure might be important in regulating the thermal acclimation of methanogenesis.” If this is the case, then why was this trend not reflected in the relative abundance of methanogens in Fig. 3b?

[Response] Many thanks for this suggestion. In the revised manuscript, the two types of soils were analyzed separately to make our results more convincing (please see Fig. 4). In addition, following the 4th comment by Reviewer #1, we expressed the magnitude of thermal response as a function of β -diversity for avoiding confusion arising from the expression as ‘relative change’.

For the relative abundance, it refers to the evenness of distribution of individuals among species in a community, and does not reflect the quantity of the microbial community and the inter-sample differences among taxa. These may be the reasons why the shifts in the relative abundance of specific methanogens could not explain the compensatory response of microbial methanogenesis to temperature change (please see pages 6-7, lines 141-151). We hope you will be satisfied with our improvements. Thanks!

Overall, we appreciate for the three reviewers' insightful comments. These comments enabled us to have much deeper thinking on the use of terms (acclimation/adaptation), data analysis/explanation, and the research implications. By addressing these comments, we feel that the revised manuscript has been greatly improved. Thank you!

References

- Atkin, O. K. & Tjoelker, M. G. Thermal acclimation and the dynamic response of plant respiration to temperature. *Trends Plant Sci.* **8**, 343-351 (2003).
- Bradford, M. A., Watts, B. W. & Davies, C. A. Thermal adaptation of heterotrophic soil respiration in laboratory microcosms. *Glob. Change Biol.* **16**, 1576-1588 (2010).
- Bradford, M. A. et al. Cross-biome patterns in soil microbial respiration predictable from evolutionary theory on thermal adaptation. *Nat. Ecol. Evol.* **3**, 223-231 (2019).
- Hartley, I. P., Hopkins, D. W., Garnett, M. H., Sommerkorn, M. & Wookey, P. A. Soil microbial respiration in arctic soil does not acclimate to temperature. *Ecol. Lett.* **11**, 1092-1100 (2008).
- Karhu, K. et al. Temperature sensitivity of soil respiration rates enhanced by microbial community response. *Nature* **513**, 81-84 (2014).
- Loveys, B. R. et al. Thermal acclimation of leaf and root respiration: an investigation comparing inherently fast- and slow-growing plant species. *Glob. Change Biol.* **9**, 895-910 (2003).
- Reich, P. B. et al. Boreal and temperate trees show strong acclimation of respiration to warming. *Nature* **531**, 633-636 (2016).

REVIEWERS' COMMENTS

Reviewer #1 (Remarks to the Author):

I am satisfied that the authors have addressed all of my comments with a high degree of competency.

Reviewer #2 (Remarks to the Author):

This is a revised version of a manuscript I previously reviewed. As best as I can tell, the authors have responded fully and made appropriate changes to the manuscript in line with the suggestions from reviewers. I have no further specific comments at this stage, except to suggest again to the authors to consider adding further detail to the discussion to provide wider context and comparison for their results.

Responses to Reviewer #1

[**Comment 1**] I am satisfied that the authors have addressed all of my comments with a high degree of competency.

[**Response**] Thank you.

Responses to Reviewer #2

[**Comment 1**] This is a revised version of a manuscript I previously reviewed. As best as I can tell, the authors have responded fully and made appropriate changes to the manuscript in line with the suggestions from reviewers. I have no further specific comments at this stage, except to suggest again to the authors to consider adding further detail to the discussion to provide wider context and comparison for their results.

[**Response**] Thank you. We further compared the results between microbial methanogenesis and CO₂ respiration in this version (please see page 6, lines 143-145). Indeed, we already highlighted the significance of the work and its potential future role in assessing greenhouse gas emission (please see page 2, lines 26-28; page 4, lines 110-111; page 6, lines 144-149, 150-157). As we pointed out in the discussion, CH₄ emissions at the ecosystem level are complex, and there are still uncertainties in the types of methanogenic compensatory response (please see page 6, lines 157-161). In order to properly interpret our research implications, we did not extend the applicability of our results to a big scale. In the discussion, we emphasized that future work should assess the potential effects of the soil environment and other biotic factors on the compensatory response of microbial CH₄ respiration over larger spatial and temporal scales, aiming to gain a better understanding of the microbial control of CH₄ emissions at the ecosystem level in a warmer world (please see page 6, lines 161-164). Hope you would comprehend our concerns.